# HALAND: Human-AI Coordination via Policy Generation from Language-guided Diffusion

## Abstract

Developing intelligent agents that can effectively coordinate with diverse human partners is a fundamental goal of artificial general intelligence. Previous approaches typically generate a variety of partners to cover human policies, and then either train a single universal agent or maintain multiple best-response (BR) policies for different partners. However, the first direction struggles with the stochastic and multimodal nature of human behaviors, and the second relies on costly few-shot adaptations during policy deployment, which is unbearable in real-world applications such as healthcare and autonomous driving. Recognizing that human partners can easily articulate their preferences or behavioral styles through natural languages and make conventions beforehand, we propose a framework for Human-AI Coordination via Policy Generation from Language-guided Diffusion, referred to as Haland. Haland first trains BR policies for various partners using reinforcement learning, and then compresses policy parameters into a single latent diffusion model, conditioned on task-relevant language derived from their behaviors. Finally, the alignment between task-relevant and natural languages is achieved to facilitate efficient human-AI coordination. Empirical evaluations across diverse cooperative environments demonstrate that Haland generates agents with significantly enhanced zero-shot coordination performance, utilizing only natural language instructions from various partners, and outperforms existing methods by approximately 89.64%.

## 1 Introduction

One of the primary objectives of artificial general intelligence is to develop intelligent agents capable of effectively coordinating with humans to achieve shared goals, known as human-AI coordination (Carroll et al., 2019; Li et al., 2024; Wang et al., 2024b;a). This holds significant potential in applications such as industrial assembly system (Nourmohammadi et al., 2022), healthcare (Gleichauf et al., 2022), video games (Siu et al., 2021), etc. Despite the impressive progess made by cooperative multi-agent reinforcement learning (MARL) in enabling agents to collaborate towards common goals across various domains (Oroojlooy & Hajinezhad, 2023), some researches apply MARL to promote human-AI coordination, but it is challenging for MARL agents to effectively coordinate with human partners (Mirsky et al., 2022; Yuan et al., 2023b). The difficulty arises because agents trained by traditional MARL find it challenging to understand the intentions and preferences of different human collaborators and fail to adapt their behaviors accordingly (Ji et al., 2023).

A canonical approach to developing the cooperative agent, often referred to as the ego agent, entails mimicking human behavior using real human data via behavioral cloning (BC) and training the best response (BR) to the fixed BC policy through reinforcement learning (RL) (Hu et al., 2022; Lou et al., 2023; Yan et al., 2024). However, this method necessitates the laborious and costly task of collecting extensive human data. Alternatively, some approaches train the ego agent without relying on human data by creating diverse partner agents beforehand, with the expectation that they can cover diverse human policies. These approaches can be broadly classified into two main directions for downstream deployment. One direction aims to train a single universal ego agent capable of effectively cooperating with various human players. Among them, self-play (SP) and other-play (OP) methods train the ego agent by repeatedly playing against a single partner, but they may become entrenched in the specific cooperative pattern (Silver et al., 2017; Hu et al., 2020). Furthermore, population-based training (PBT) methods (Long et al., 2023) first generate a diverse pool of partners by maximizing the divergence of trajectory distribution (Lupu et al., 2021), population (Zhao et al.,

2023), or minimizing cross-play rewards (Charakorn et al., 2022), etc. Subsequently, they train a common best response ego agent to adapt to different partners. In contrast to learning an universal ego agent, the other direction trains a group of ego agents or augments the ego agent policy with auxiliary partner identifier, and selects the appropriate one through techniques such as few-shot adaptation when faced with unknown human partners. For instance, Maze coevolves two populations of ego agents and partners and selects the most suitable policy during testing (Xue et al., 2022a), while Macop develops high-compatibility cooperative training paradigms by continuously expanding policy heads (Yuan et al., 2023a), showing stronger coordination ability in complex scenarios.

However, existing methods in these two directions have certain limitations. Firstly, developing an universal policy requires meticulous design to ensure efficient training and suffers from stochastic and multimodal human behaviors due to limited model capacity (Wang et al., 2024b). Secondly, the few-shot adaptation process for partner identification or policy selection typically requires to run multiple episodes beforehand, which can be costly and even unachievable in real-world scenarios, such as medical application (Coronato et al., 2020) and automatic driving (Yan et al., 2022). These limitations hinders the development of efficient human-AI coordination. Besides, neither direction showcases explainability towards human preference explicitly. Note that humans often reach conventions (Shih et al., 2021; Guan et al., 2023) before coordination by expressing their own behavior styles or preferences with language. Therefore, a natural question arises: *Can we achieve efficient human-AI coordination with language instructions only?*

To tackle the above issues, we propose Haland, an efficient human-AI coordination framework via policy generation from language-guided diffusion. Concretely, given a set of diverse partners, we first train corresponding BR policies to each partner via reinforcement learning. Inspired by the powerful expressiveness and generation capability of Latent Diffusion Model (LDM) (Rombach et al., 2022), we compress the parameters of these BR policies into a single generative model conditioning on task-relevant language derived from their behaviors, so as to deal with stochastic and multimodal human behavior. Afterwards, we achieve alignment between the task-relevant and natural languages by introducing a tailored language translator. During deployment, a human partner provides the language instruction with respect to preference and it will be translated into corresponding task-relevant language, Haland then generates the ego agent policy that can effectively coordinate with the human partner through the conditional denoising process. We demonstrate Haland's superior collaborative capabilities across various human-AI coordination environments, including both single-task and multi-task settings with diverse partners.

## 2 RELATED WORK

**Human-AI Coordination** endeavors to empower AI systems with the capabilities of effectively coordinating with diverse human partners (Dafoe et al., 2021; Yuan et al., 2023b; Wang et al., 2024b). One direction is to model human behaviors from real human data via behavioral cloning (BC). However, high-quality human data is costly to collect in real-world scenarios beforehand. Alternatively, existing works on human-AI coordination without human data can be broadly categorized into two main directions. They both create diverse partners in the hope that human policies during testing can be covered. The first direction is to train a single universal ego agent to coordinate with diverse partners. Among the plethora of methods, self-play (SP) approaches (Tesauro, 1994; Silver et al., 2017) involve training ego agents by coordinating against themselves, while other-play (Hu et al., 2020) introduces diversity into coordination patterns to disrupt the symmetry of self-play policies. Population-based training methods have emerged as prevalent approaches to enhance policy diversity. For instance, FCP (Strouse et al., 2021) introduces diversity by employing different random seeds and checkpoints at various training stages. MEP (Zhao et al., 2023) and TrajeDi (Lupu et al., 2021) optimize population-level entropy objectives alongside coordination returns to achieve a diverse population. The other direction trains a group of ego agents or augments one ego agent with auxiliary partner identifier. For instance, Maze coevolves two populations of ego agents and partners through evolution (Xue et al., 2022a). However, these methods require the human partner to coordinate with probing policies for a few episodes, to select the proper ego agent or attain the correct identifier.

**Language-guided Reinforcement Learning** involves training agents to perform tasks based on Natural Language (NL) instructions (Luketina et al., 2019). Previous methods focus on training instruction-following agents by exposing NL instructions to RL policies directly. For instance, litera-

ture (Hill et al., 2020) encodes NL instructions using a pre-trained language model and incorporates the NL embedding into the policy. Literature (Chaplot et al., 2018) combines human instructions with agent observations using a multiplication-based mechanism and pre-trains the instruction-following policy through behavior cloning (Pomerleau, 1991). Alternatively, literature (Akakzia et al., 2021) encodes NL instructions into a manually-designed binary vector where each element represents specific semantics. The concept of instruction-following policies also has connections with Hierarchical RL (Barto & Mahadevan, 2003), where NL instructions naturally serve as task abstractions for low-level policies (Blukis et al., 2021). HAL (Jiang et al., 2019) leverages the compositional structure of NL to make decisions directly at the NL level for solving long-term, complex RL tasks. Furthermore, TALAR (Pang et al., 2023) introduces task-related task languages as a unique representation of NL instructions that is easily interpretable by the policy. Instead of directly exposing NL instructions to policies, Haland reconstructs cooperative policies through guided diffusion generation with translated NL instructions, more related work are discussed in App. A.

## 3 BACKGROUND

**Two-player Cooperative Markov Game**   Most human-AI coordination problems can be modeled as a two-player cooperative Markov Game (Littman, 1994), which is described by a tuple $\langle \mathcal{S}, \mathcal{A}_1, \mathcal{A}_2, \mathcal{T}, R \rangle$. $\mathcal{S}$ is the set of states, $\mathcal{A}_1$ and $\mathcal{A}_2$ are the action spaces of the two agents, respectively, which can be different in a heterogeneous setting. $\mathcal{T} : \mathcal{S} \times \mathcal{A}_1 \times \mathcal{A}_2 \times \mathcal{S} \to [0, 1]$ is the transition function, and the joint action of two agents result in a shared reward given by $R : \mathcal{S} \times \mathcal{A}_1 \times \mathcal{A}_2 \to \mathbb{R}$. At each time step, agents receive the state $s_t$ and output actions $a_t^1 \in \mathcal{A}^1, a_t^2 \in \mathcal{A}^2$. The joint action leads to the next state $s_{t+1} \sim \mathcal{T}(\cdot|s_t, a_t^1, a_t^2)$ and a global reward $R(s_t, a_t^1, a_t^2)$.

**Diffusion Model**   The diffusion models are a category of generative models by modeling the process of synthetic data as thermodynamic diffusion process. The remarkable success in various domains has showcased its powerful generation capability and has been used in RL for planning or functioning as expressive policies recently (Yang et al., 2023).

For each training datapoint $\mathbf{x}_0 \sim p_{\text{data}}(\mathbf{x})$, diffusion models construct a Markov chain $\mathbf{x}_0, \mathbf{x}_1, ..., \mathbf{x}_N$ in the forward process by adding noise with pre-defined noise scales $0 < \beta_1, .., \beta_N < 1$, such that $p(\mathbf{x}_i|\mathbf{x}_{i-1}) := \mathcal{N}(\mathbf{x}_i; \sqrt{1 - \beta_i}\mathbf{x}_{i-1}, \beta_i \mathbf{I})$. It can be further derived that $p(\mathbf{x}_i|\mathbf{x}_0) = \mathcal{N}(\mathbf{x}_i; \sqrt{\bar{\alpha}_i}\mathbf{x}_0, (1 - \bar{\alpha}_i)\mathbf{I})$, where $\alpha_i = (1 - \beta_i), \bar{\alpha}_i = \prod_{j=1}^{i} \alpha_j$. The noise scales are chosen such that $\mathbf{x}_N \sim \mathcal{N}(\mathbf{0}, \mathbf{I})$. In the reverse diffusion process, the samples can be generated by starting from $\mathbf{x}_N \sim \mathcal{N}(\mathbf{0}, \mathbf{I})$ and following the recursion:

$$\mathbf{x}_{i-1} = \frac{1}{\sqrt{\alpha_i}}(\mathbf{x}_i - \frac{1 - \alpha_i}{\sqrt{1 - \bar{\alpha}_i}}\boldsymbol{\epsilon}) + \sqrt{\beta_i}\mathbf{z}, \tag{1}$$

where $\boldsymbol{\epsilon} \sim \mathcal{N}(\mathbf{0}, \mathbf{I})$ is the noise added during the re-parameterization of forward process $\mathbf{x}_i = \sqrt{\bar{\alpha}_i}\mathbf{x}_0 + \sqrt{1 - \bar{\alpha}_i}\boldsymbol{\epsilon}$ and $\mathbf{z}$ is a sample from the standard normal distribution. To predict the noise, the denoising network $\epsilon_\theta$ is instantiated and optimized through through the following objective:

$$\mathcal{L}_{\text{denoise}} = \sum_{i=1}^{N} \mathbb{E}_{\mathbf{x}_0 \sim p_{\text{data}}(\mathbf{x}), \epsilon \sim \mathcal{N}(\mathbf{0}, \mathbf{I})}[||\boldsymbol{\epsilon} - \epsilon_\theta(\sqrt{\bar{\alpha}_i}\mathbf{x}_0 + \sqrt{1 - \bar{\alpha}_i}\boldsymbol{\epsilon}, i)||^2]. \tag{2}$$

## 4 METHOD

In this section, we introduce our proposed method, Haland, an efficient Human-AI Coordination framework via Language-guided Diffusion, which generates cooperative policies based on natural language instructions provided by users (see Fig. 1). The problem settings and formulations will be introduced in Sec. 4.1. Next, the details of cooperative policy compression and techniques for language alignment are discussed in Sec. 4.2 and Sec. 4.3, respectively. Finally, an overall pipeline for cooperative policy compression and language-guided policy generation is presented in Sec. 4.4.

### 4.1 PROBLEM FORMULATION

Humans often reach conventions (Guan et al., 2023) before coordination by expressing their own behavior styles or preference with natural language (NL) instructions, but existing works fail to utilize

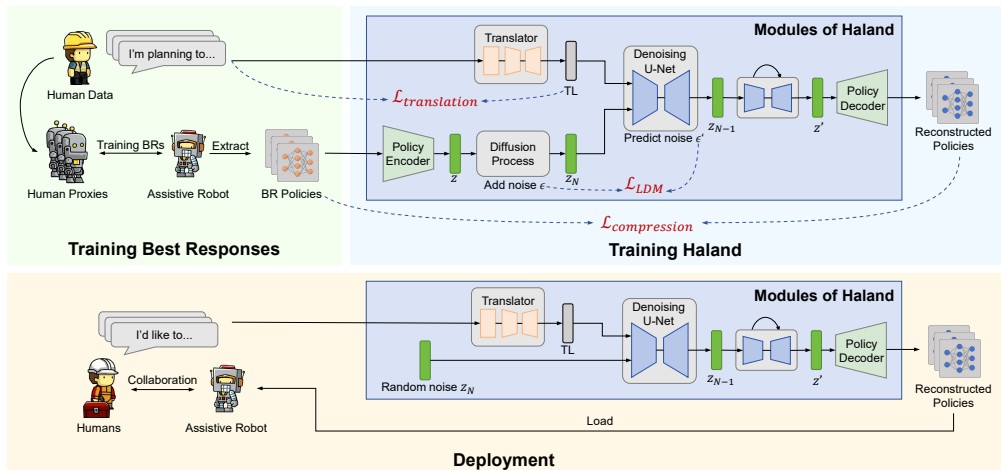

Figure 1: Architecture of Haland. Given human proxies and corresponding language instructions, we first train the respective best response (BR) polices. Subsequently, during training, Haland learns to translate natural language (NL) instructions into task language (TL) embeddings and compress the BR policies into a single diffusion model. During deployment, given a NL instruction, Haland first transforms NL instruction into TL embedding and then reconstructs the BR policy accordingly.

the abundant information in the NL. In our work, we take the instructions from human partners into consideration, and extend the standard two-player cooperative Markov Game into the NL-guided version by introducing natural language instructions. The NL-guided two-player cooperative Markov Game can be formalized as $\langle \mathcal{S}, \mathcal{A}_1, \mathcal{A}_2, \mathbf{L}_N, \mathcal{T}, R \rangle$, where $\mathbf{L}_N$ is the natural language instruction space, and the other elements hold the same meanings.

During training, a set of diverse human partner policies $\{\pi_P^j\}_{j=1}^{N_H}$ paired with NL instructions $\{\{L_N^{j,k} \in \mathbf{L}_N\}_{k=1}^K\}_{j=1}^{N_H}$ are provided, where $N_H$ is the number of partner policies, $K$ is the number of the natural language instructions for each partner policy. Here $\{L_N^{j,k}\}_{k=1}^K$ share similar semantic meanings and differ only in expressions considering the variability of natural language. To solve the human-AI coordination problem with NL instructions, we aim to train the ego agent policy $\pi_E(\cdot|s, L_N)$ to maximize the following objective:

$$\mathcal{J}(\pi_E) = \mathbb{E}_{L_N}\left[\sum_t \mathbb{E}_{s_t, a_t^1 \sim \pi_E(\cdot|s_t, L_N), a_t^2 \sim \pi_P(\cdot|s_t)}[R(s_t, a_t^1, a_t^2)]\right], \tag{3}$$

where $L_N$ is the NL instruction representing the behavior styles or preference of the unknown human player $\pi_P$ during deployment.

## 4.2 POLICY GENERATION FROM LANGUAGE-GUIDED DIFFUSION

To achieve human-AI coordination when provided with a set of diverse training partners $\{\pi_P^j\}_{j=1}^{N_H}$, traditional approaches are broadly categorized into two directions. One direction aims to train a single universal ego agent while the other trains a group of best response policies. However, they either suffer from stochastic and multimodal human behaviors (Pearce et al., 2022) or require costly few-shot adaptation. To benefit from the both directions while overcoming the limitations, we propose to distill the multiple best response policies into a single NL-guided diffusion model for policy generation, due to its powerful generation capability. Similar approach was first proposed in literature (Hegde et al., 2023), which distills the quality-diversity policy archive into the diffusion model conditioning on behavior descriptions.

Specifically, we first train the best response policies $\{\pi_{\text{BR}}\}_{j=1}^{N_H}$ to the given partners and expect to compress them into one single diffusion model. However, the complex and variable structures of neural networks make it difficult to directly conduct diffusion process on parameter space, we then compress policy parameters into a compact latent space, named policy representation space, using the variational autoencoder (VAE) $f = (f_{\mathcal{E}}, f_{\mathcal{D}})$. In specific, we assume that each best response

policy $\pi_{\text{BR}}$ is instantiated by a Multi-Layer Perceptron (MLP) comprising $M$ layers, where each layer containing a weight matrix $W_m$ and bias vector $b_m$, $1 \leq m \leq M$. We encode the weight matrix $W_m$ and bias vector $b_m$ into latent embeddings $z = f_{\mathcal{E}}(\pi_{\text{BR}}) \in \mathbb{R}^d$ using the convolutional neural network (CNN) and MLP, respectively. To reconstruct the policy $\hat{\pi}_{\text{BR}}$, the decoder $f_{\mathcal{D}}$ incorporates a conditional graph hypernetwork (Hegde & Sukhatme, 2023), which estimates the parameters of the policy network by taking the latent representation $z$ as input. By reconstructing policy action distribution instead of parameters, the encoder and decoder are jointly trained for policy compression:

$$\mathcal{L}_{\text{compress}}(f_{\mathcal{E}}, f_{\mathcal{D}}) = \sum_{j=1}^{N_H} \mathbb{E}_{s \sim \mathcal{D}}[\text{dist}(\hat{\pi}_{\text{BR}}^j(\cdot|s), \pi_{\text{BR}}^j(\cdot|s))] + \mathcal{D}_{\text{KL}}\left[ f_{\mathcal{E}}(z|\pi_{\text{BR}}^j) \| \mathcal{N}(\mathbf{0}, \mathbf{I}) \right], \quad (4)$$

where $\mathcal{D}$ is the replay buffer, $\text{dist}(\cdot, \cdot)$ measures the discrepancy between two action distributions, $\mathcal{D}_{\text{KL}}$ is the Kullback-Leibler (KL) divergence and $\hat{\pi}_{\text{BR}}^j = f_{\mathcal{D}}(f_{\mathcal{E}}(\pi_{\text{BR}}^j))$.

With the trained VAE capable of compressing and reconstructing policy parameters, we now attain the expressive and compact latent space of best response policies and are capable of training the latent diffusion model (LDM) on such space. Our goal is to generate appropriate policy representation given the natural language instruction $L_N \in \mathbf{L}_N$ expressing the partner's preference. To accomplish this, we directly train a conditional diffusion model $\mathcal{M} = (\epsilon_\theta, \tau_\theta)$ conditioning on $L_N$. Formally, provided with the policy latent representations $\{z^j = f_{\mathcal{E}}(\pi_{\text{BR}}^j)\}_{j=1}^{N_H}$ and language instructions $\{\{L_N^{j,k}\}_{k=1}^K\}_{j=1}^{N_H}$, the diffusion model is trained following the standard latent diffusion training objective:

$$\mathcal{L}_{\text{LDM}}(\epsilon_\theta, \tau_\theta, \{\{L_N^{j,k}\}_{k=1}^K\}_{j=1}^{N_H}) = \sum_{j=1}^{N_H} \sum_{i=1}^{N} \sum_{k=1}^{K} \mathbb{E}_{z^j \sim f_{\mathcal{E}}(\pi_{\text{BR}}^j), \epsilon \sim \mathcal{N}(\mathbf{0}, \mathbf{I})} \left[ \|\epsilon - \epsilon_\theta(z_i^j, i, \tau_\theta(L_N^{j,k})\|_2^2 \right],$$
$$(5)$$

where $z_i^j = \sqrt{\bar{\alpha}_i} z^j + \sqrt{1 - \bar{\alpha}_i}\epsilon$ as introduced in Eqn. 2. During deployment, given the language instruction $L_N$ provided by human collaborators, a policy representation $\hat{z}_0$ is sampled by starting with Gaussian noise $\hat{z}_N$ and refining $\hat{z}_i$ into $\hat{z}_{i-1}$ at each diffusion timestep with the perturbed noise $\epsilon_\theta(z_i, i, \tau_\theta(L_N))$ following Eqn. 1. Subsequently, we recover the policy with the trained decoder $\hat{\pi}_{\text{BR}} = f_{\mathcal{D}}(\hat{z}_0)$, with the hope that it could effectively coordinate the human partner. Detail training and deployment pipeline will be discussed in Sec. 4.4.

### 4.3 LANGUAGE ALIGNMENT FOR ROBUST GENERATION

Although language-guided diffusion enables the generation of cooperative policies aligned with partner preferences, the variability and redundancy inherent in natural language present several challenges for the training and deployment of diffusion model. First, natural language exhibits variability and humans may have different linguistic conventions, which means a specific instruction can be conveyed using different expressions. Consequently, if the generator is trained using only a limited set of expressions, it may struggle to effectively generate cooperative policies when confronted with unfamiliar expressions during real-world deployment. Second, natural language instructions often contain syntactic components irrelevant to specific tasks. These redundant syntactic components, combined with the variability of natural language, pose challenges for the diffusion model in aligning various natural language instructions with corresponding policy representations.

To address the challenges posed by the variability and redundancy of natural language, we develope a suite of task-relevant language to accurately capture task-specific information that can also accurately reflect the behavior styles of policies during training. Specifically, in our developed task language (TL), each policy is associated with a unique TL embedding $L_T^j$ which can be learnable or event-based embeddings, facilitating clear differentiation between different policies by the diffusion model. Subsequently, we construct a translator to map the variable natural language (NL) instructions $\{L_N^{j,k}\}_{k=1}^K$ with similar semantic meanings to unique TL embeddings $L_T^j$, so as to ensure the zero-shot coordination with natural language instructions only. Detailed information on the design of TL can be found in App. D.

Concretely, the translator is composed of a pre-trained Bert (Devlin et al., 2019) model and a VAE $g = (g_{\mathcal{E}}, g_{\mathcal{D}})$. Given a NL instruction $L_N$, we first encode it via the Bert model $\mathcal{B}$ and obtain the embedding $b$. Subsequently, the encoder of the VAE processes $b$ to produce an intermediate representation $e = g_{\mathcal{E}}(b)$, which is then used to recover the task language (TL) embedding $\tilde{L}_T =$

$g_{\mathcal{D}}(e)$, the encoder and decoder are optimized based on the following standard VAE objective:

$$\mathcal{L}_{\text{translation}}(g_{\mathcal{E}}, g_{\mathcal{D}}) = \sum_{j=1}^{N_H} \sum_{k=1}^{K} ||\tilde{L}_T^{j,k} - L_T^j||_2^2 + \mathcal{D}_{\text{KL}}[g_{\mathcal{E}}(e|\mathcal{B}(L_N^{j,k}))||\mathcal{N}(\mathbf{0}, \mathbf{I})], \qquad (6)$$

where $\tilde{L}_T^{j,k} = g_{\mathcal{D}}(g_{\mathcal{E}}(\mathcal{B}(L_N^{j,k})))$. Since the natural instructions used in coordination can be notably different from the broader internet data for Bert pre-training, we fine-tune the model via predicting the teammate's behavioral type to achieve domain adaptation. This process also enhances the model's ability to capture semantic similarities, which in turn facilitates the translation from NL to TL. Specifically, the Bert model is encapsulated into a classifier $\mathcal{C}$ and fine-tuned via minimizing the cross-entropy objective:

$$\mathcal{L}_{\text{finetune}}(\mathcal{B}, \mathcal{C}) = -\frac{1}{N_H} \sum_{j=1}^{N_H} \frac{1}{K} \sum_{k=1}^{K} \log P\left[\mathcal{C}(L_N^{j,k}) = j\right]. \qquad (7)$$

### 4.4 OVERALL TRAINING AND DEPLOYMENT PROCESS

We here provide the overall description of the procedure of our approach Haland. A detailed description of the overall architecture can be found in App. B. During the training phase, we first train the best response policies $\{\pi_{\text{BR}}^j\}_{j=1}^{N_H}$ given diverse partners $\{\pi_P^j\}_{j=1}^{N_H}$. Afterwards, we compress the policy parameters into a expressive and compact latent space by training VAE $f = (f_{\mathcal{E}}, f_{\mathcal{D}})$ based on Eqn. 4. Subsequently, the latent diffusion model $\mathcal{M} = (\epsilon_\theta, \tau_\theta)$ is optimized to recover policy representations conditioning on TL embeddings. We replace $\{\{L_N^{j,k}\}_{k=1}^{K}\}_{j=1}^{N_H}$ in Eqn. 5 into $\{L_T^j\}_{j=1}^{N_H}$ considering the variability and redundancy inherent in natural language. Finally, the translator between NL instructions and TL embeddings, which comprises a pre-trained Bert model and a VAE, is optimized based on Eqn. 6 and Eqn. 7.

During the deployment phase, when presented with a natural language instruction $L_N$ from the unknown human partner, Haland first converts $L_N$ into TL embedding $\tilde{L}_T$ with the trained translator. Subsequently, the latent diffusion model generates appropriate policy representation $\hat{z}_0$ via denoising sampling steps conditioning on $\tilde{L}_T$. Finally, the policy is reconstructed by decoder: $\hat{\pi} = f_{\mathcal{D}}(\hat{z}_0)$.

## 5 EXPERIMENTS

In this section, we conduct extensive experiments on multiple two-player cooperative environments to answer the following questions: 1) Can Haland achieve superior coordination performance with diverse partners compared to baselines across various scenarios (Sec. 5.2)? 2) Is the population-based training paradigm capable of producing robust ego agents when faced with collaborators exhibiting diverse high-level behaviors (Sec. 5.2)? 3) Does Haland demonstrate robustness to the the variabilty and redundancy of natural language instructions provided by unknown partners (Sec. 5.3)? 4) Whether Haland is capable of coordinating with novel human partners? 5) How do different components of Haland influence its coordination performance (Sec. 5.5)?

### 5.1 ENVIRONMENTS AND BASELINES

We consider multiple environments (Fig. 2), where **Overcooked** (Carroll et al., 2019) is a fully-observable two-player cooperative cooking environment, where two agents work together to prepare and serve soup to obtain shared rewards. In order to obtain training and evaluation partners with diverse behavioral styles and preferences, we design four novel layouts which yield multiple collaborative solutions, including *Center Pots*, *Crossway*, *Diverse Coordination* and *Diverse Orders*. Specifically, in *Diverse Orders*, the agents are required to prepare soup with specific types of ingredient and serve the soup to target serving spot. In other layouts, agents exhibit diverse behaviors by considering different preferences, including the position of ingredients or serving spots. **Level-Based Foraging(LBF)** (Papoudakis et al., 2021) is a partially observable grid world game, where agents and foods are assigned different levels. A group of agents can collect the food only if all of them choose the loading action and the summation of their levels is greater than the level of food. We

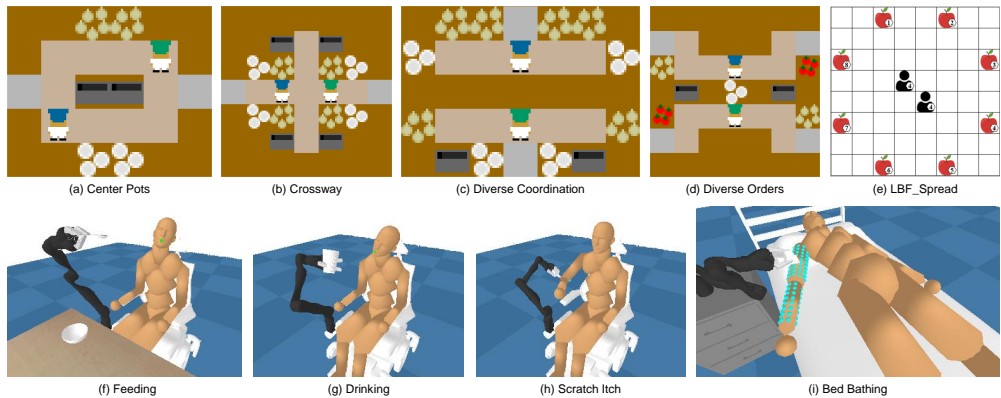

Figure 2: Experimental environments used in this paper.

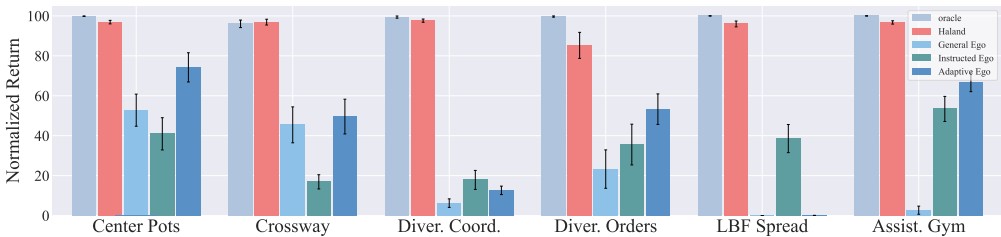

Figure 3: Comparison with baslines in multiple environments. Here Diver. Coord., Diver. Orders and Assist. Gym are abbreviations for Diverse Coordination, Diverse Orders and Assistive Gym.

designed the *LBF Spread* layout, where eight food with different levels are uniformly distributed along the edges. The ego agent needs to identify the target food by observing the partner's behaviors or relying on external instructions. And **Assistive Gym** (Erickson et al., 2020) is a physics-based simulation framework designed for physical human-robot interaction and robotic assistance, featuring continuous action and observation spaces. We select an assistive robot, Jaco, and four assisting tasks to demonstrate HALAN's capability of providing assistance in a multi-task setting.

For baselines, we consider different training settings and policy architectures aimed at developing a robust ego agent or a group of ego agents capable of accommodating diverse partners through population-based training, including: 1) **General Ego** trains an ego agent with a diverse population of partners. 2) **Instruction-Following Ego** trains an ego agent with the partner population, incorporating partners' labels as part of the ego's input, also noted as **Instructed Ego**. 3) **Adaptive Ego** trains an ego agent with the partner population, incorporating partners' actions as part of the ego's input.

In each environment, we constructed a set of teammates with different behavioral styles. We set aside half of generated teammates for evaluation and the other half for training. The best response policies for each partner serve as the **Oracle** for comparisons. More details about the training of diverse partners and different baselines can be found in App. C.2 and App. C.3, respectively. TL is defined by the frequency of high-level events, and the details of design is discussed in App. D

## 5.2 RESULTS ANALYSIS

**Coordination Performance** At first glance, we compare Haland against the mentioned baselines to investigate the coordination ability with the diverse partner population, as shown in Fig. 3. The results are normalized and averaged over different cooperative partners and 5 random seeds. When only training an universal policy, General Ego performs poorly in most layouts and even collapses in *LBF Spread*, validating that a common best response can suffer from diverse behaviors of partners. Instructed Ego augments the General Ego agent with partner label in hope to discriminate multimodal behaviors of partners, and achieves adequate performance improvement. By incorporating the partner's action into the input of the universal ego agent, Adaptive Ego outperforms Instructed

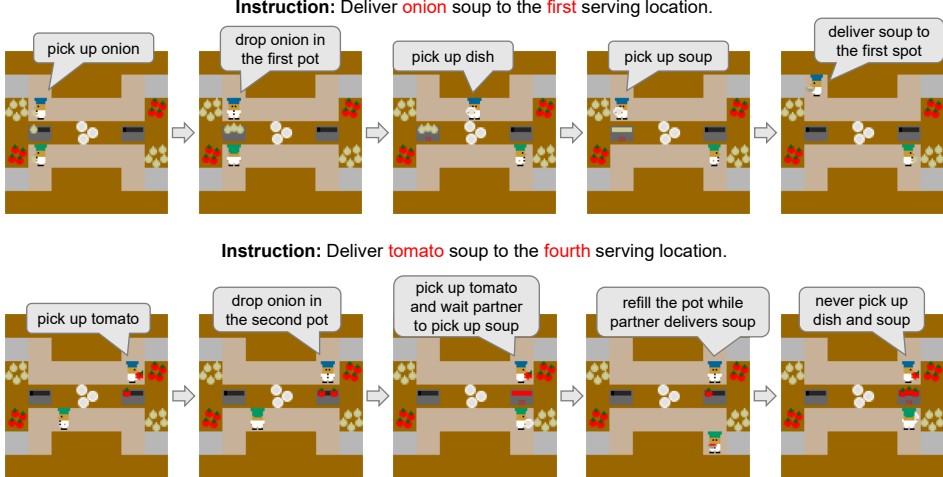

Figure 5: Two demonstrations of the coordination process in the *Diverse Orders* layout. The ego agents are generated via Haland guided by the NL instructions of the partner agents.

Ego as it implicitly performs teammate modeling, while Instructed Ego ignores information in NL instructions, verifying that the agent benefits more from task-relevant information than instruction only. However, none of them demonstrates coordination performance of a common best response across all partners. Haland achieves the best overall coordination performance on all benchmarks and is comparable to the Oracle, showing the effectiveness and high efficiency of the proposed method.

**Population-based Training Analysis** To investigate why population-based training paradigm that trains an universal ego agent fails to coordinate well with partners of diverse behavior styles. We demonstrate the detailed coordination performance of the strongest baseline, Adaptive Ego, in Fig. 4, where the oracle coordination performance with 8 partners in *Crossway* layout is presented as benchmark. We can find each Adaptive Ego agent trained with different random seeds can adapt to different portions of partners, five out of eight at most, but struggle to effectively coordinate with the others compared to Oracle. This underscores the difficulty in accommodating the multimodal behaviors of partners, even when training with a diverse population. Instead, Haland fully utilizes the expressiveness and multimodal modeling capability of diffusion model, successfully dealing with the multimodal behavior challenge by recovering the corresponding best response policies through natural language instructions only.

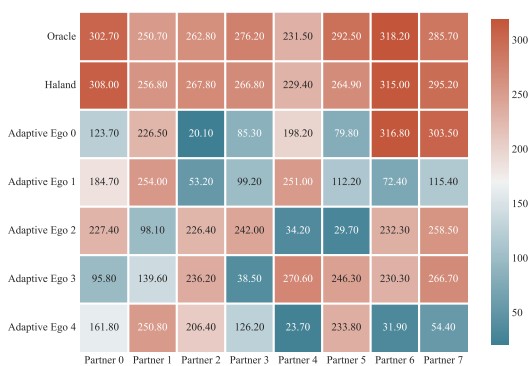

Figure 4: PBT Analysis: Pairwise coordination results of Adaptive Egos on *Crossway* layout.

**Coordination Visualization** To verify whether the agents developed by Haland understand the natural language instructions provided by the partner, we visualize the different coordination processes during deployment in the *Diverse Orders* layout. As shown in Fig. 5, agents receiving the natural language instructions, which indicate the partner's behavioral style or preference, perform corresponding skills to achieve effective coordination. The text boxes highlight the specific skills exhibited by the ego agent. For instance, when coordinating with the partner who prefers to deliver onion soup instead of tomato soup to the first serving location, the ego agent actively picks onions to cook soup and serve it to the exact position, fulfilling the requirements in the NL instructions. More demonstrations in the *Diverse Orders* layout can be found in App. G.

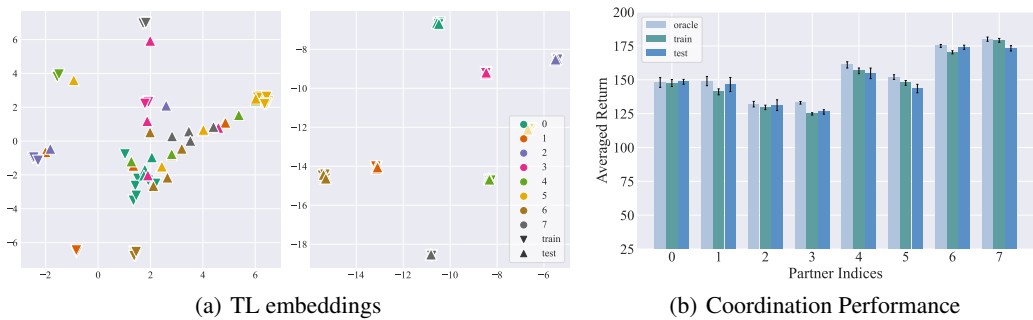

(a) TL embeddings          (b) Coordination Performance

Figure 6: (a) TL embeddings converted from NL instructions before and after fine-tuning the Bert model. (b) Coordination performance of generated ego agents after fine-tuning the Bert model.

## 5.3 LANGUAGE GENERALIZATION

**Language Embedding** To highlight the necessity of fine-tuning the Bert model , we compare the embedding results of NL instructions before and after fine-tuning the Bert model using t-SNE (Van der Maaten & Hinton, 2008). The training instruction set consists NL instructions conveying similar meanings but featuring various expressions, which is translated into TL to training the diffusion model. The evaluation instruction set comprises instructions with similar meanings to those in the training set but expressed differently. As shown in Fig. 6(a), before fine-tuning the Bert model with the sequence classification task, the translator is limited to converting NL instructions from the training set into similar TL embeddings, while NL instructions from the testing set cannot align well with the TL embeddings. However, after fine-tuning the Bert model, the translator exhibits the capability to convert diverse NL instructions into aligned TL embeddings, even when encountering NL instructions with similar semantics that were not seen during its training phase.

**Generalization Performance** Fig. 6(b) demonstrates HALAN's generalization ability over different NL instructions. Both the training and testing NL instruction sets are translated into TL embeddings using the translator with the fine-tuned Bert model. During training, only the TL embeddings from the training set are available to the diffusion model. We can find that the NL instructions from both the training and testing sets guide the diffusion model to accurately reconstruct ego agents with high collaborative capability. Detailed examples of NL instructions can be found in App. F.

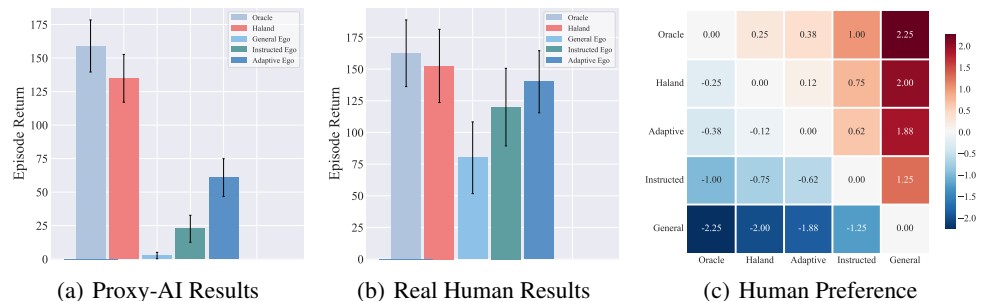

(a) Proxy-AI Results     (b) Real Human Results     (c) Human Preference

Figure 7: Human-AI experiments in the *Diverse Orders* layout. (a) Collaborative performance with behavior-cloned human proxies. (b) Collaborative performance with real human players. (c) Human preference scores for the row partner compared to the column partner.

## 5.4 HUMAN EVALUATION

Our ultimate goal is to develop agents capable of coordinating with novel human partners. In this section, we conducted an online study to evaluate agents generated by Haland and baselines in collaborative play with both human proxies and 8 real human partners. The proxies and participants

Table 1: Ablation results in the *Diverse Coordination* layout. Numbers $0 \sim 7$ are partner indices.

| Partner | Oracle | HALAN | *W/o Diff-MLP* | *W/o Diff-UNet* | *W/o Translator* | *W/o VAE* |
|---|---|---|---|---|---|---|
| 0 | $155.2 \pm 24.39$ | $\mathbf{159.0 \pm 7.7}$ | $145.2 \pm 35.1$ | $0.0 \pm 0.0$ | $0.0 \pm 0.0$ | $0.0 \pm 0.0$ |
| 1 | $\mathbf{147.2 \pm 40.63}$ | $133.2 \pm 47.4$ | $135.0 \pm 43.8$ | $5.0 \pm 10.7$ | $0.0 \pm 0.0$ | $0.0 \pm 0.0$ |
| 2 | $130.4 \pm 30.0$ | $\mathbf{131.2 \pm 19.5}$ | $47.0 \pm 23.0$ | $0.0 \pm 0.0$ | $0.0 \pm 0.0$ | $0.0 \pm 0.0$ |
| 3 | $128.0 \pm 13.3$ | $\mathbf{130.0 \pm 11.8}$ | $112.4 \pm 22.3$ | $0.0 \pm 0.0$ | $0.0 \pm 0.0$ | $0.0 \pm 0.0$ |
| 4 | $\mathbf{163.2 \pm 34.8}$ | $162.4 \pm 15.4$ | $139.2 \pm 44.9$ | $4.0 \pm 8.0$ | $0.0 \pm 0.0$ | $0.0 \pm 0.0$ |
| 5 | $142.0 \pm 32.8$ | $\mathbf{146.2 \pm 19.1}$ | $0.0 \pm 0.0$ | $0.0 \pm 0.0$ | $0.0 \pm 0.0$ | $0.0 \pm 0.0$ |
| 6 | $169.4 \pm 24.9$ | $159.2 \pm 36.0$ | $172.0 \pm 11.7$ | $29.2 \pm 28.6$ | $\mathbf{174.2 \pm 18.0}$ | $0.5 \pm 3.1$ |
| 7 | $180.0 \pm 18.9$ | $182.0 \pm 15.4$ | $181.0 \pm 14.8$ | $178.0 \pm 16.6$ | $\mathbf{186.4 \pm 15.6}$ | $11.2 \pm 17.8$ |

are required to coordinate with a full cohort of agents in the *Diverse Orders* layout. As shown in Fig. 7(a), when coordinating with human proxies constructed via behavior cloning from human-play trajectories, the performance of baseline agents significantly declined compared to the results when coordinating with held-out partners. Nevertheless, Haland demonstrated collaborative performance on par with the Oracle, highlighting the effectiveness of tailored collaborative policy generation guided by language instructions.

As illustrated in Fig. 7(b), Haland continued to achieve a level of collaborative performance comparable to that of the Oracle even when coordinating with real human players. Furthermore, we calculated pairwise differences in average ratings to derive preference values, revealing that participants expressed a clear preference for the collaborative policies generated by Haland over baselines except for the Oracle, as shown in Fig. 7(c). More experimental details could be found in App. H.

## 5.5 ABLATION STUDY

Haland is composed of multiple components, and we conducted ablation studies in the *Diverse Coordination* layout to investigate their impacts. First, to highlight the outstanding conditional generation capability of the latent diffusion model $\mathcal{M}$, we replaced it with a Conditional Adversarial Generative Model (CGAN) (Mirza & Osindero, 2014) conditioning on the partner labels. We implemented two variations of CGAN: one using an MLP network and another utilizing the same UNet structure as the diffusion model, denoted as *W/o Diff-MLP* and *W/o Diff-UNet*, respectively. Next, to emphasize the importance of the task language and the translator for language alignment, we removed the translator and directly train the diffusion model conditioning on NL instructions, denoted as *W/o Translator*. Finally, for policy compression and reconstruction, we derived *W/o VAE* by removing the VAE $f = (f_{\mathcal{E}}, f_{\mathcal{D}})$ and attempted to directly distill best response policies using a diffusion model. As shown in Tab. 1, CGAN fails to model all best response policies effectively. The replacement of MLP with UNet also fails to enhance the generation capability of the CGAN. Furthermore, after removing the translator for language alignment, the diffusion model collapses to generating only two best response policies due to the high similarity between NL instructions. After removing the VAE in the latent diffusion model, directly modeling the distribution of policy parameters is challenging for diffusion model as *W/o VAE* shows in the table.

## 6 CONCLUSION AND FUTURE WORK

This paper tackles the challenge of zero-shot human-AI coordination during deployment, harnessing the intuitive nature of human expression through natural language instructions. We introduce a novel framework, named Human-AI Coordination via Policy Generation from Language-guided Diffusion (Haland), which compresses diverse best response policies into a single diffusion-based generator. Empirical evaluations conducted across diverse cooperative environments validate the effectiveness of Haland. Haland enhances policy deployment through language instruction, moving away from few-shot adaptation. In the future, it could be extended to address policy shifts using techniques like detection and adaptation in open machine learning settings (Zhou, 2022), where teammates may experience policy changes within a single episode (Zhang et al., 2023). Additionally, combining this approach with stronger language models such as T5 (Raffel et al., 2020) for real-world embodied tasks (Liu et al., 2024b) holds significant potential.

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

## A  DETAILED RELATED WORK

**Multi-Agent Reinforcement Learning (MARL)**    (Albrecht et al., 2023) involves a team of agents learning a joint policy to tackle tasks through interactions with the environment, optimizing their policies through reinforcement learning (Sutton & Barto, 2018). Compared to traditional methods, MARL offers advantages in handling environmental uncertainty and learning to solve unknown tasks without excessive domain knowledge. However, MARL introduces new challenges distinct from single-agent settings. On one hand, in environments where multiple agents coexist, observations are often partially observable, limiting individual agents' access to global information and hindering optimal decision-making (Zhu et al., 2022). On the other hand, as multiple agents learn simultaneously, policies dynamically change, rendering the environment non-stationary from an individual agent's perspective, which may impede convergence (Papoudakis et al., 2019). Moreover, in cooperative MARL scenarios where agents share common goals, the challenge of accurately assigning rewards to facilitate efficient cooperation learning, known as credit assignment, becomes crucial (Yuan et al., 2023b; Wang et al., 2021a). Additionally, as the number of agents in a Multi-Agent System (MAS) (Dorri et al., 2018) increases, the search space for solving RL problems exponentially expands, posing scalability issues and making policy learning and search extremely challenging (Zhang, 2011; Christianos et al., 2021). In recent years, the fusion of deep learning

and MARL has yielded significant advancements, with various algorithms proposed and applied to address complex tasks. These include policy gradient-based methods like MADDPG (Lowe et al., 2017) and MAPPO (Yu et al., 2022), value-based methods such as VDN (Sunehag et al., 2018) and QMIX (Rashid et al., 2018), and approaches leveraging transformer architectures to enhance coordination capabilities, like MAT (Wen et al., 2022). Meanwhile, MARL has garnered widespread attention and showcased significant progress across various fields (Zhang et al., 2021), demonstrating promising applications in path planning (Chung et al., 2024), autonomous driving (Zhang et al., 2024b), active voltage control (Wang et al., 2021b), and dynamic algorithm configuration (Xue et al., 2022b).

**Human-AI Coordination**   endeavors to empower AI systems with the capabilities needed for cooperation and to nurture collaboration (Dafoe et al., 2020; 2021). Considerable attention has been directed toward the development of AI systems or agents adept at effectively coordinating with diverse human collaborators in recent years. A prevalent approach involves techniques such as modeling (Albrecht & Stone, 2018) to understand others' intentions or behavior, or constructing an effective behavior model over human data and planning with this model (Sheridan, 2016). However, these methods often entail an expensive and time-consuming data-collection process. Building on the achievements of cooperative MARL (Yuan et al., 2023b), numerous related approaches like ad-hoc teamwork (AHT) (Mirsky et al., 2022), few-shot teamwork (FST) (Fosong et al., 2022), and zero-shot coordination (ZSC) (Treutlein et al., 2021) have emerged. AHT tackles the challenge of designing agents capable of coordinating with new teammates without prior coordination (Stone et al., 2010). In FST settings (Fosong et al., 2022), agents trained within a team to accomplish one task are combined with agents from different tasks, requiring them to adapt collectively to an unseen but related task. In ZSC settings, ego agents are trained to interact with various partners during the training phase, enabling successful coordination with novel partners or human collaborators, garnering significant attention across different domains. Among the plethora of methods, self-play (SP) approaches (Tesauro, 1994; Silver et al., 2017) involve training ego agents by competing against themselves, while other-play (Hu et al., 2020) introduces diversity into coordination patterns by training agents with another agent, disrupting the symmetry of self-play policies. Population-based methods have emerged as prevalent approaches to enhance policy diversity. For instance, FCP (Strouse et al., 2021) introduces diversity by employing different random seeds and checkpoints at various training stages. MEP (Zhao et al., 2023) and TrajeDi (Lupu et al., 2021) optimize population-level entropy objectives alongside coordination returns to achieve a diverse population. These methods operate under the premise that exposing the ego agent to training partners with diverse skills, preferences, and behavioral styles enhances its robustness and enables collaboration with novel partners. However, they often yield agent populations with only low-level or policy-level diversity, overlooking the multimodal challenge associated with adapting a single ego agent to partners with diverse high-level behavioral styles, preferences, and skills. Alternatively, MAZE (Xue et al., 2022a) maintains separate ego agent and partner populations and trains both simultaneously through coevolution, while ensemble approaches are necessary to determine the optimal cooperative action during deployment. Macop develops high-compatibility cooperative training paradigms by continuously expanding policy heads (Yuan et al., 2023a). These methods represent a further step toward effective coordination with diverse and multimodal teammates. Others focus on open-ended coordination (Li et al., 2023b), biased human (Yu et al., 2023), open ad hoc teamwork (Wang et al., 2024a), human-AI coordination evaluation (Wang et al., 2024b), combining with LLM (Li et al., 2023a; Liu et al., 2024a), etc.

**Language-guided Reinforcement Learning**   involves training agents to perform tasks based on Natural Language (NL) instructions (Luketina et al., 2019). Previous methods focus on training instruction-following agents by exposing NL instructions to RL policies directly. For instance, Literature (Hill et al., 2020) encodes NL instructions using a pre-trained language model and incorporates the NL encoding into the policy. Literature (Chaplot et al., 2018) combines human instructions with agent observations using a multiplication-based mechanism and pre-trains the instruction-following policy through behavior cloning (Pomerleau, 1991). Alternatively, Literature (Akakzia et al., 2021) encodes NL instructions into a manually-designed binary vector where each element represents specific semantics. The concept of instruction-following policies has connections with Hierarchical RL (Barto & Mahadevan, 2003), where NL instructions naturally serve as task abstractions for low-level policies (Blukis et al., 2021). HAL (Jiang et al., 2019) leverages the compositional structure of NL to make decisions directly at the NL level for solving long-term, complex RL tasks. Furthermore,

TALAR (Pang et al., 2023) introduces a task-related task language as a unique representation of NL instructions that is easily interpretable by the policy. Instead of directly exposing NL instructions to policies, Haland reconstructs cooperative policies aligned with the requirements specified in NL instructions through language-guided diffusion. In the Human-AI setting, Literature (Hu & Sadigh, 2023) develops InstructQ and InstructPPO that enables humans to specify what kind of strategies they expect from their AI partners through natural language instructions. Proagent (Zhang et al., 2024a)harnesses large language models (LLMs) to create proactive agents capable of dynamically adapting their behavior to enhance cooperation with teammates. SAMA (Li et al., 2023a) proposes a novel "disentangled" decisionmaking method, Semantically Aligned task decomposition in MARL (SAMA), that prompts pre-trained language models with chain-of-thought that can suggest potential goal for efficient coordination. HAPLAN (Guan et al., 2023) ask humans to give their preferences to the LLM and review the proposed conventions, ensuring an effective human-AI coordination with a better alignment to human biases. One recent work HLA (Liu et al., 2024a) also employs LLM to facilitate human-AI coordination. However, its main idea is to build an instruction-following agent with LLM, requiring a continuous stream of natural language instructions from human. This places the whole burden on human and can lead to inefficient coordination.

## B  OVERALL ARCHITECTURE OF HALAN

Fig. 1 illustrates the overall architecture of our approach, Haland, which comprises three primary stages: data preparation, training, and deployment. During the data preparation phase, human data, including behavioral or preference data, along with corresponding NL descriptions or instructions, are collected. These collected human data are then utilized to construct human proxies through techniques such as imitation learning behavioral cloning (BC). During the subsequent training phase, we begin by training the best response (BR) policies using the constructed human proxies. Then, the modules of Haland, which include the VAE for policy compression and reconstruction, the diffusion model for policy generation, and the translator for language alignment, are trained. During the deployment phase, NL instructions are converted into TL embeddings using the trained translator. Subsequently, the latent diffusion model generates appropriate policy representations conditioning on these TL embeddings. Not that during the training phase, $z_N$ is produced via the diffusion process, which involves iteratively adding noise to the policy latent representation $z$. Whereas, in the deployment phase, $z_N$ is directly obtained through sampling from the Gaussian distribution.

## C  EXPERIMENT SETTINGS

### C.1  ENVIRONMENTS

**Overcooked**    In the Overcooked (Carroll et al., 2019) environment, two players are placed into a grid-world kitchen as chefs and tasked with preparing and delivering as many soups cooked with required ingredients as possible in limited time. To successfully deliver a dish, the agents need to collaborate to accomplish a sequence of sub-tasks, including collecting ingredients, depositing ingredients into cooking pots, turn on the cooking pots, collecting dishes and getting the cooked soup, and delivering the soup to the delivering location. The soup will take twenty seconds to cook and the agent will receive a reward of twenty after succesfully delivering a soup. For simple usage of the Overcooked environment and compatibility with the Stable-Baslines3, we utilize the open-source framework PantheonRL[1](Sarkar et al., 2022).

In order to produce partner populations with diverse high-level behavioral styles and preferences, we design four novel layouts which yield multiple coordination patterns: 1) *Center Pots* layout includes two cooking pots located in the center of the kitchen, surrounded by a ring-shaped one-way passage. The agents can collaborate in a left-right manner, where each agent works on a single side independently, or in a up-down manner, where one agent focuses on collecting, depositing and cooking onions, the other agent focuses on collecting dishes, getting the cooked soup and deliver it to the serving location. The roles of two agents in both coordination manners can exchange, which enables a multitude of possible coordination patterns. 2) *Crossway* is another shared-space layout involving a narrow one-way crossing. Both the cooking pots and delivering locations are located

---

[1]https://github.com/Stanford-ILIAD/PantheonRL

at the end of passages. In order to avoid blocking each other, the agents need to collaborate in a delicate manner and adapting to each other's movements. There are two sets of cooking pots and two delivering locations, which also yields multiple coordination patterns. 3) *Diverse Coordination* layout involves two separated rooms, each agent is located in one room. The whole kitchen is left-right symmetric and both agents can deliver the soup, while only the partner agent (Green) is able to cooking soup. Detailed usage and possible coordination patterns will be discussed in Sec. C.2. 4) *Diverse Orders* is another separated-space layout and used as a multi-task layout. There are two types of ingredients and four delivering locations, while to accomplish a specific task, the agents need to cook the soup with a specified type of ingredients and deliver the soup to a specified serving location.

**LBF**   We designed the fully-observable and fully-cooperative *LBF Spread* layout, where eight foods with different levels are uniformly distributed along the edges. *LBF Spread* is also used as a multi-task fully-cooperative environment. In *LBF Spread*, the food can only be collected with two agents together, and for each task a specific target food level is given. The ego agent need to identify the target food by observing the partner's behaviors or relying on an external instruction, and the agents will receive a reward of 1 only after the target food is collected.

**Assistive Gym**   Assistive Gym (Erickson et al., 2020) is a physics-based simulation framework designed for physical human-robot interaction and robotic assistance, featuring continuous action and observation spaces. This simulation framework models various activities of daily living (ADLs): itch scratching, drinking, feeding, body manipulation, dressing and bathing. Assistive Gym also models a person's physical capabilities and preference for assistance, which are used to provide a reward function. Due to the simulation of realistic human movement, training a policy in Assistive Gym is particular time-consuming. Training a PPO policy in Stable-Baselines3 (SB3) (Raffin et al., 2021) will take more than 4 days with a 36 vCPU machine. To this end, we select an assistive robot, Jaco, and four assisting tasks and use them jointly as a multi-task environment, without considering different impairments and preferences of the human, to reduce the time consumption for training policies. In all tasks, the impairment level is set to *None* and the preference of human is set to the default values.

## C.2   RULES FOR CONSTRUCTING DIVERSE PARTNER POPULATION

To develop partners with diverse high-level behavioral styles, we manually designed a set of rules to constrain the partners' skills or achievable locations in the *Overcooked* environment. In each layout, we produced a set of eight diverse partners following these rules. For each behavioral style, we trained the partner using 10 different seeds, allocating half for training Haland and the other half for evaluation.

**Center Pots**   In the *Center Pots* layout, two cooking pots are located in the center of the kitchen, surrounded by a ring-shaped passage. We obtain agent pairs with diverse coordination patterns by limiting the working space of the partner agent. These constraints include working only on the upper/lower side, only on the left/right side, or only in a specific corner, like the upper left side of the kitchen. (1) When the partner agent works only on the left side, the best coordination strategy for the ego agent is to work on the right side independently. (2) When the partner agent works only on the right side, the pattern is symmetric to the previous pattern. (3) When the partner agent works only on the upper side, focusing on tasks related to the onions, the best coordination strategy for the ego agent is to work on the lower side and focus on tasks related to soup delivery. (4) When the partner agent works only on the lower side, the roles of the ego agent and the partner agent switch, symmetric to the previous pattern. (5)-(8) When the partner agent works only in a specific corner, such as the upper left side, focusing on tasks related to onions and only working with the left cooking pot, the best coordination strategy for the ego agent is to focus on complementary tasks for the same cooking pot.

**Crossway**   In the *Crossway* layout, agents work in a shared narrow crossing, requiring them to carefully adapt their behaviors to each other. We obtain agent pairs with diverse coordination patterns by limiting the partner agent's movement and skills within the crossing. (1)-(4) When the partner agent works only with cooking pots on the upper/lower side and delivery locations on the left/right side, the optimal coordination strategy for the ego agent is to work with cooking pots and delivery locations on the opposite sides to avoid blocking each other. (5)-(6) When the partner agent works

only with cooking pots on the upper/lower side and cannot collect dishes, the ego agent is responsible for collecting dishes, getting the soup, and delivering the soup. (7)-(8) When the partner agent focuses only on downstream tasks related to soup delivery and uses only the left/right delivery location, the ego agent is responsible for collecting onions and handling cooking tasks.

**Diverse Coordination**  In the *Diverse Coordination* layout, the ego agent and the partner agent operate in separate spaces. The partner agent has access to all the resources necessary to fulfill an order, but we introduce diversity in partners' styles and preferences by limiting their movements and skills. The scenarios are as follows: (1)-(2) When the partner agent works only on the left side and cannot collect onions or dishes from the dispensers, the ego agent must work on the same side and pass onions or dishes through the counter to the partner agent. (3) When the partner agent works only on the left side and cannot collect both onions and dishes from the dispensers, the ego agent must pass both resources through the counter on the left side. (4)-(6) When the partner agent works only on the right side, three additional patterns emerge, symmetric to the first three. (7)-(8) When the partner agent works only on the left or right side and the delivery location at the bottom is unavailable, the partner agent must pass the cooked soup through the counter, and the ego agent is responsible for relaying and delivering the soup to the delivery location at the top.

**Diverse Orders**  The *Diverse Orders* layout is designed as a multi-task environment, featuring four different delivery locations and two different ingredients. For each task, the agents must prepare soup with a specified ingredient (onion or tomato) and deliver the cooked soup to a specified delivery location, resulting in a total of eight distinct tasks.

## C.3  BASELINES

We leverage Proximal Policy Optimization (PPO) (Schulman et al., 2017) from Stable-Baselines3 (SB3) (Raffin et al., 2021) as the training algorithm for both ego agents and partner agents. In particular, we utilize recurrent value and policy networks [2] comprising Long Short-Term Memory (LSTM) (Hochreiter & Schmidhuber, 1997) units with a hidden size of 256 to enhance the adaptability of universal egos. Conversely, the policy networks for partner agents consist of simple Multi-Layer Perceptrons (MLPs).

**Oracle**  Ego agents trained alongside the diverse partners are approximations of the best responses and serve as the Oracle policies specific to each partner.

**General Ego**  This approach involves training a single ego agent with a diverse population of partners, expecting it to accommodate different partners based solely on observation.

**Instruction-Following (Instructed) Ego**  This approach is similar to the General Ego, with the distinction that during training, the policy input comprises one-hot labels indicating different partners for collaboration. The Instruction-Following Ego's policy integrates an instruction-embedding module, implemented as MLP, to process the partner labels.

**Adaptive Ego**  This approach is similar to the Instruction-Following Ego, yet it differs in incorporating the partners' one-hot actions during collaboration, rather than partner labels, as part of the policy input. By taking the partners' actions as input, Adaptive Ego implicitly performs teammate modeling.

## C.4  COMPUTE RESOURCES

We run our experiments on GeForce RTX 2080 Ti. For training diverse partners with SB3, a typical training of $6 \times 10^5$ steps takes approximately an hour in the Overcooked environment. In the Assistive Gym environment, the training requires more than $1 \times 10^7$ steps to obtain usable policies, which takes around $2 \sim 4$ days using 36 concurrent simulation actors. For the components of Haland, training the VAE and the diffusion model takes only $1 \sim 2$ hours, whereas training the translator takes about 10 hours due to the incorporation of the Bert model. In the deployment phase, the inference time of the diffusion model for policy generation is negligible ($< 1$ second).

---

[2] https://github.com/Stable-Baselines-Team/stable-baselines3-contrib

## D  DESIGN OF TASK LANGUAGE

In our work, we define a set of $U$ high-level task-relevant events and represent task-relevant descriptions using the frequencies of these events in the Overcooked environment, denoted as $\mathbf{v} \in \mathbb{R}^U$. To stabilize the training of the language-conditioned diffusion model, we normalize each dimension by dividing it by the largest frequency to obtain $\hat{\mathbf{v}}$. We then discretize the values in each dimension into $V$ tokens using the technique proposed in literature (Dong et al., 2024) as follows:

$$L_T^u = \lfloor \text{clip}(\hat{v}^u, 0, 1 - \delta) \cdot V \rfloor + (u - 1)V, \quad u = 1, \cdots, U \tag{8}$$

where $\delta$ is a small slack variable, and we set $V = 10$ in this work. Since the entire set of predefined possible events is large, resulting in sparse task-relevant descriptions of partners, we discard events that have zero values across all partners. Fig. 8, Fig. 9, Fig. 10, and Fig. 11 show the normalized statistics of high-level task-relevant events in each layout, which are used to construct the TL embeddings of diverse partners. In these heatmaps, deeper colors indicate higher frequencies of high-level events. For the LBF and Assistive Gym environments, we use sinusoidal encodings of task labels as the TL embeddings.

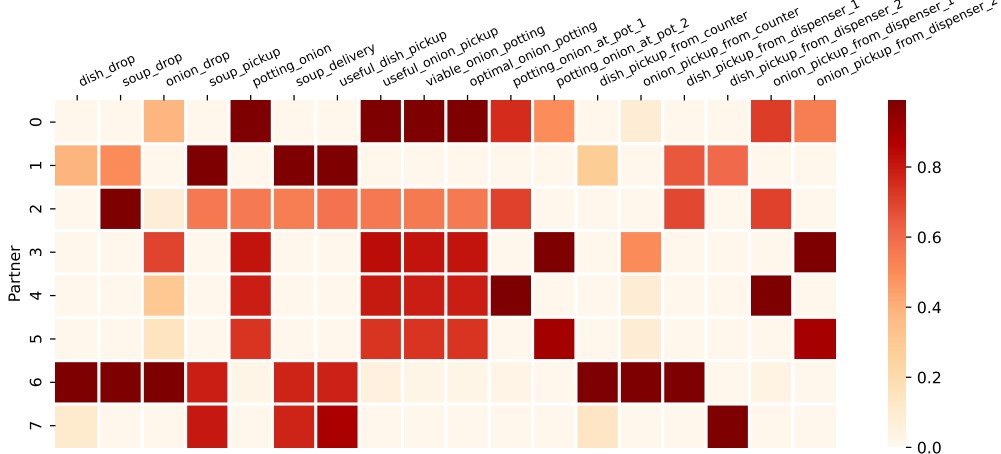

Figure 8: Heatmap of diverse partners' high-level behaviors in the *Center Pots* layout.

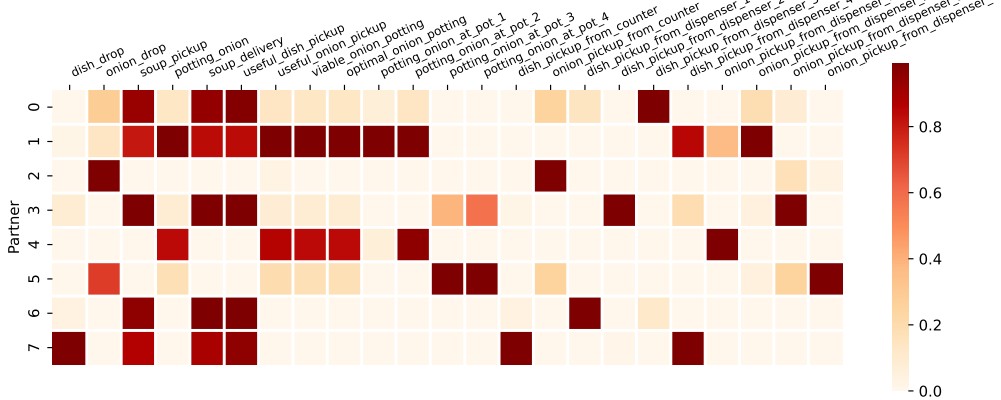

Figure 9: Heatmap of the diverse partners' high-level behaviors in the *Crossway* layout.

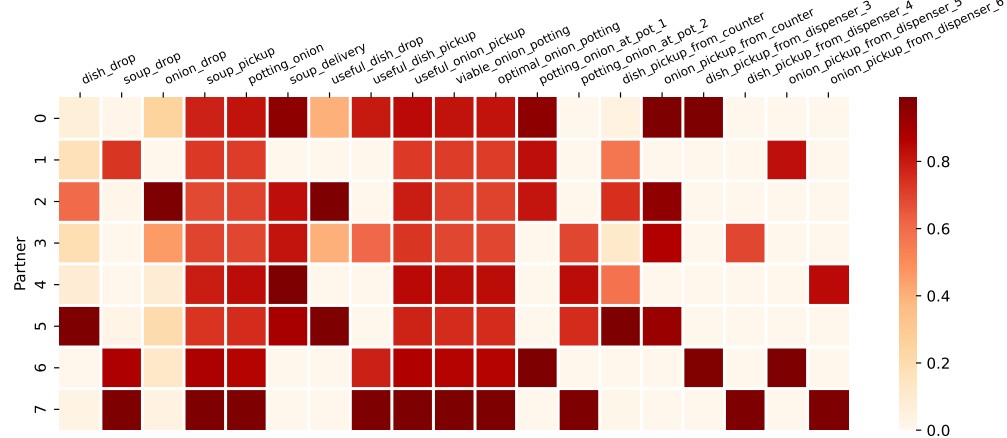

Figure 10: Heatmap of the diverse partners' high-level behaviors in the *Diverse Coordination* layout.

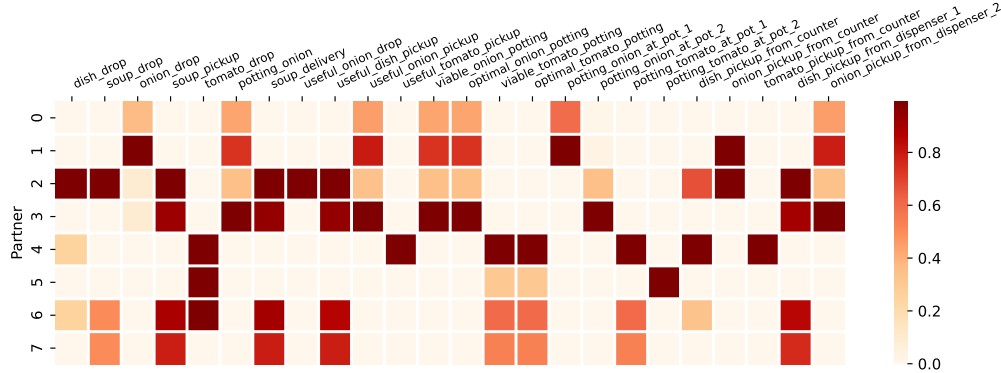

Figure 11: Heatmap of the diverse partners' high-level behaviors in the *Diverse Orders* layout.

# E IMPLEMENTATION DETAILS OF HALAN

Haland involves a Variational AutoEncoder for policy compression, a Latent Diffusion Model for language-guided generation and a Translator for language alignment.

## E.1 VARIATIONAL AUTOENCODER

The Variational Autoencoder (VAE) comprises an encoder designed to compress policy parameters into low-dimensional representations, coupled with a conditional Graph Hypernetwork (GHN) responsible for estimating the policy parameters based on these representations. The encoder incorporates $M$ Conv1D blocks to process the weight matrices $W_m$ and an MLP to process the bias vectors $b_m$, where $m = 1, \cdots, M$ and $M$ is the number of layers in the policy network. Following this, a final fully connected layer processes the concatenated features and generates the latent representations. The architecture of the encoder is derived from the structure proposed in literature (Hegde et al., 2023), while the GHN implementation is adapted from literature (Hegde & Sukhatme, 2023).

## E.2 LATENT DIFFUSION MODEL

In line with the approach outlined in literature (Hegde et al., 2023), we employ a UNet backbone as the architecture for the Latent Diffusion Model (LDM). As suggested by literature (Hegde et al., 2023), we integrate a spatial transformer into the Attention Module, enabling cross-attention between intermediate features and language embeddings. Further details regarding the Residual Block and

Table 2: Hyperparameters for the VAE

| Name | Value |
|---|---|
| Latent Representation Dimension | 64 |
| Encoder Hidden Dimension | 64 |
| KL Coefficient | 1e-6 |
| Gradient Clipping | True |
| Learning Rate | 1e-4 |
| GHN Hidden Layer Size | 16 |

Attention Block can be explored in the provided open-source implementation[3]. It's worth noting that minor adjustments to the architecture have been observed to have minimal impact on performance in practical applications. The architecture of the UNet used in our work is presented in Tab. 3.

Table 3: Architecture of the UNet

| Module | Submodules |
|---|---|
| Encoder | Conv2D $\times$ 2 + Positional Embedding + Residual Block |
| Bottleneck | Residual Block $\times$ 2 + Attention Block |
| Decoder | Residual Block $\times$ 2 + Attention Block + Upsample |

### E.3 TRANSLATOR

The translator incorporates a fine-tuned Bert model and a VAE incorporating MLPs.

Table 4: Hyperparameters for the Translator

| Name | Value |
|---|---|
| Bert max sequence length | 32 |
| Bert context dimension | 768 |
| VAE encoder network | $[1024, 1024, 64]$ |
| VAE decoder network | $[1024, 1024, |L_T|]$ |
| Learning Rate | 1e-4 |

## F NL GENERALIZATION EXAMPLES

As illustrated in Fig. 6(b), the Translator utilizing the fine-tuned Bert model demonstrates impressive generalization capabilities. Specifically, we utilize ten diverse NL descriptions to characterize the behavioral styles and preferences of each partner in the training set, along with an additional four distinct descriptions for the testing set. For example, in the *Diverse Coordination* layout, the NL descriptions used in the training and testing sets for *Partner_0* and *Partner_1* are detailed as follows:

**Traning set**

- *Partner_0*:
  - "Focus on tasks on the left side and avoid handling onions from the onion dispenser."
  - "Perform duties on the left side only and avoid interacting with the onion dispenser."
  - "Engage in tasks on the left and do not handle onions from the onion dispenser."
  - "Stick to responsibilities on the left and ignore the onion dispenser."

---

[3]https://github.com/hkproj/pytorch-stable-diffusion

- – "Limit actions to the left side and avoid interacting with the onion supply."
- – "Perform tasks on the left without involving the onion dispenser."
- – "Focus on left-side responsibilities, excluding tasks related to the onion dispenser."
- – "Concentrate on left-side duties and refrain from engaging with the onion dispenser."
- – "Concentrate on left-side activities and avoid collecting onions from the onion dispenser."
- – "Work specifically on the left side of the kitchen and refrain from picking up onions from the dispenser."

- *Partner_1*:
    - – "Focus on tasks on the left side and avoid handling dishes from the dish dispenser."
    - – "Perform duties on the left side only and avoid interacting with the dish dispenser."
    - – "Engage in tasks on the left and do not handle dishes from the dish dispenser."
    - – "Stick to responsibilities on the left and ignore the dish dispenser."
    - – "Limit actions to the left side and avoid interacting with the dish supply."
    - – "Perform tasks on the left without involving the dish dispenser."
    - – "Focus on left-side responsibilities, excluding tasks related to the dish dispenser."
    - – "Concentrate on left-side duties and refrain from engaging with the dish dispenser."
    - – "Concentrate on left-side activities and avoid collecting dishes from the dish dispenser."
    - – "Work specifically on the left side of the kitchen and refrain from picking up dishes from the dispenser."

**Testing set**

- *Partner_0*:
    - – "Concentrate on tasks allocated for the left side of the kitchen, refraining from any interaction with the onion dispenser."
    - – "Execute tasks on the left side exclusively, avoiding any engagement with the onion supply."
    - – "Stick to assigned responsibilities on the left side and abstain from handling onions from the dispenser."
    - – "Focus solely on tasks pertaining to the left side, ensuring no involvement with the onion dispenser."

- *Partner_1*:
    - – "Dedicate efforts to tasks designated for the left side of the kitchen, refraining from handling dishes from the dispenser."
    - – "Concentrate solely on activities on the left side, avoiding any interaction with the dish dispenser."
    - – "Stick to responsibilities assigned for the left side and abstain from picking up dishes from the dispenser."
    - – "Focus exclusively on tasks related to the left side, ensuring no involvement with the dish dispenser."

# G   ADDITIONAL DEMONSTRATIONS

The demonstrations for the remaining six tasks in the *Diverse Orders* layout are presented in Fig. 12.

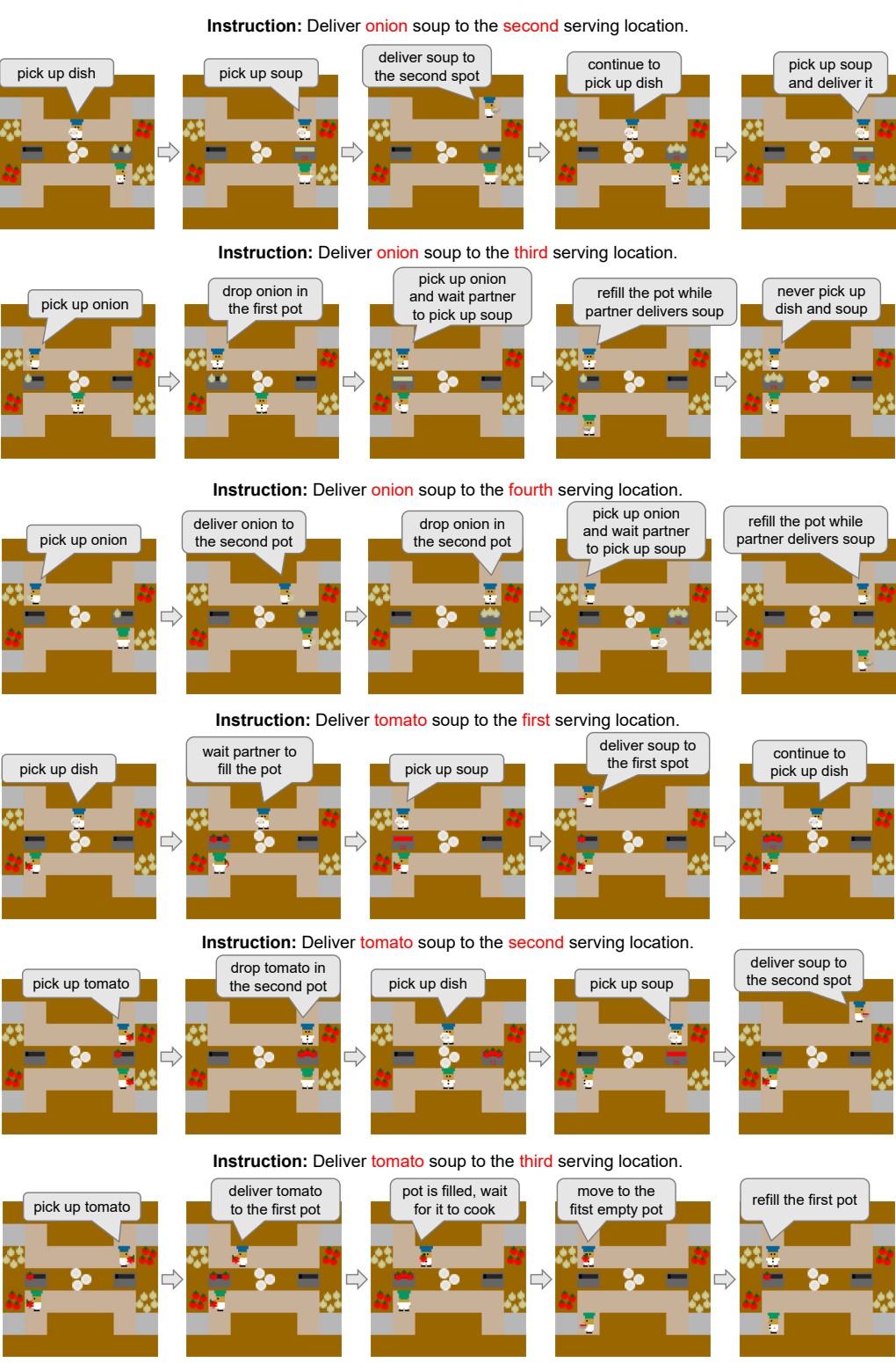

Figure 12: Demonstrations of the coordination process in the *Diverse Orders* layout, corresponding to six tasks not previously mentioned. The ego agents are generated via Haland guided by the NL instructions of the partner agents.

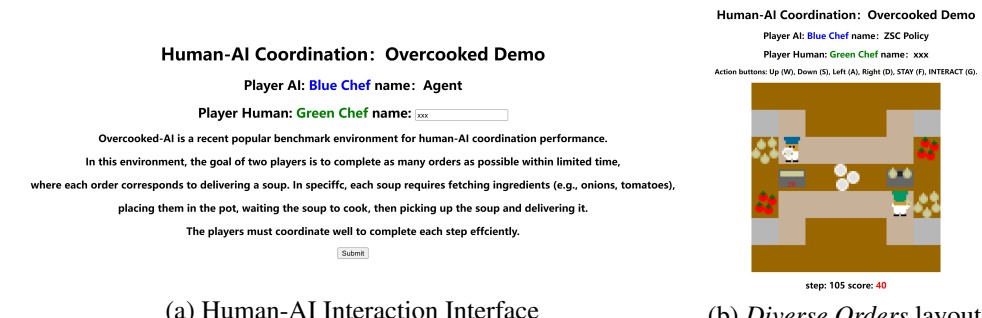

(a) Human-AI Interaction Interface

(b) *Diverse Orders* layout

Figure 13: (a) We implemented a simple web-based human-AI interaction interface for the Over-cooked environment using Flask, where humans can play with AI models using keyboard controls. (b) The human-AI interaction interface on the *Diverse Orders* layout, where the task is coordinating to deliver a specific type of soup to a designated serve location.

## H    HUMAN-AI EXPERIMENTS

To evaluate how effectively the Haland and baseline agents collaborate with real human players, we recruited eight participants for a human-AI collaboration study designed within a human-in-the-loop framework in the *Diverse Orders* layout. Participants began by reading the game instructions and viewing gameplay demonstrations. They were then tasked with collaborating as effectively as possible with the AI partner to complete as many orders as they could. Following this, participants interacted with the AI partners through a web-based interface using keyboard controls, as shown in Fig. 13. Before each round, participants provided a natural language description to specify their behavioral preferences, which is used for the selection of Oracle policy and the policy generation of Haland.

After completing each game episode (400 timesteps), participants rated their satisfaction with the AI partner on a five-point Likert-like scale, ranging from 1 (very dissatisfied) to 5 (very satisfied). Upon finishing all games involving human players and AI agents, both the collaborative performance and the human ratings for the different agents were statistically analyzed. Fig. 7(b) shows the collaborative performance of AI agents with real human players. The pairwise differences in average ratings were calculated as the human preference values, as shown in Fig. 7(c).

Additionally, human proxies exhibiting different behavioral styles were constructed through behavior cloning from human play trajectories collected using the aforementioned human-AI interaction interface in the human-in-the-loop experiments. The collaborative performance with these human proxies is presented in Figure 7(a). The primary distinction between real human players and human proxies is that human players can actively adapt to the AI agents, while the behavior of human proxies is relatively fixed. This lack of adaptability leads to a decline in performance in the Proxy-AI results compared to the results obtained with real human players.

