# OpenReview forum: "Haland: Human-AI Coordination via Policy Generation from Language-guided Diffusion"
_ICLR.cc/2025/Conference — Submitted to ICLR 2025_

### Official Review · Reviewer_ydcX · 2024-10-27

**Soundness:** 2
**Presentation:** 1
**Contribution:** 2
**Rating:** 5
**Confidence:** 3

**Summary:**

The paper tackles the issue of human-ai coordination across a diverse set of humans. The authors assert that existing solutions generally take two shapes

1. Training a single general universal agent, capable of coordinating across diverse human partners
2. Training multiple specialized "best response" (BR), each tailored to particular human partners.

The authors argue that the first faces issues with the multimodality of human behaviour, while the second is unlikely to scale to the real world due to the costly adaptations necessary.

Instead, the authors propose leveraging humans' ability to use natural language to express their preferences and behaviours ahead of time as a superior alternative for human-ai coordination.

Concretely, to do so, the authors contribute a novel human-AI coordination framework, Haland. Here, given a set of diverse partners, corresponding BR policies are trained via RL. The policy parameters are then compressed into a representational space using a conditional diffusion model. During training, the diffusion model is conditioned on "task language" representations of natural language instructions from the human partners, translated from natural language to "task language" by a translator module, built on top of BERT. At inference time, human partners can input OOD natural language descriptions of the tasks shown during training alongside a random noise vector for the compressed BR representation, and haland will output the parameters for the best response policy for the task.

After outlining the above method, the authors conduct a series of experiments in the Overcooked, Level-Based Foraging (LBF) and Assistive Gym environment, comparing Haland to a number of baselines throughout, demonstrating Haland's superiority. As part of the experimentation, the authors also study Haland's robustness to instruction prompt variability and round the section off with some ablation studies on the architecture.

**Strengths:**

The paper is somewhat original in pointing out that natural language might be effectively leveraged for improving cooperative problems, particularly when humans are involved such as in human-ai cooperation. The haland pipeline presented by the authors is impressive in how many different ingredients it elegantly ties together. The paper is generally thorough within individual experiments: I appreciated the variety of environments and the presence of ablation studies on the architecture. In general, achieving such high human-ai coordination performance _zero shot_ appears to be significant.

**Weaknesses:**

I unfortunately found the paper quite sloppy. This not only makes the paper hard to read, but also makes me as a reviewer uncertain about the claims made within. The sloppiness is particularly apparent in section 4.2. The section starts of with a repetitive paragraph that has already been stated several times before then. There are then several phrasing mistakes, such at line 213-215 "However, [...], we then" reads awkwardly. When speaking about encoding the weight matrix and bias vector of the MLP, you mention "_the_ CNN and MLP", but it is unclear which CNN you are referring to, should this have said "a" rather than "the"? The same applies to the "MLP" mentioned in that sentence. You then speak about reconstructing policy action distribution as if it had already been introduced, but the surrounding text talks about reconstructing the parameters, which are conflicting assertions, making this rather confusing. There is also mention of "the" replay buffer as if it had been introduced, but again this is only mentioned once with uncertainty for the reader about its significance. In the same paragraph, it is said that the diffusion model is conditioned and trained on natural language, but in Figure 1 and the surrounding text it is said to be conditioned on the task language, so once again we as readers are presented with conflicting information.

Other places where the presentation suffers is presenting the questions you wish to answer at the beginning of section 5, but not reminding readers of which question is being answered at each subsection, requiring the reader to jump up and down. In fact some of these questions are never directly addressed again after being introduced.

Other minor examples of sloppiness include things such as mispelling Haland as HALAN and being inconsistent with capitalization. It is a bit jarring and doesn't inspire confidence, so I would recommend a few more read throughs and revisions from the authors.

I also found it awkward that you refer to the 2nd baseline as Instruction-Following Ego and Instructed Ego. It would have been simpler to just use a single term. In general the baselines felt underdefined in the main text, and perhaps not realistically representative of existing alternatives in the literature. I would have appreciated stronger justification for the baselines chosen. On the topic of baselines paper is also lacking an explanation and/or more thorough comparison with Hegde et al., 2023 which is mentioned in passing but never expanded.

Aside from the sloppiness, I unfortunately also found the paper somewhat underdeveloped. I believe the paper would have benefited from a more honest discussion of Haland's limitations, which are currently lacking from the main text. For instance, is the VAE responsible for encoding the BRs only capable of encoding MLPs adhering to specific dimensionality? Or can it handle any incoming BR architecture? Is the robustness to variability in natural language limited to variability within the same semantic meaning since the semantic meaning of the tasks is the same across training and testing? How much can Haland generalize beyond the training BRs? These are some of the questions I found underdeveloped and/or underaddressed in the paper.

Finally, while I appreciated the existence of ablation experiments, I found these somewhat disappointing. It's particularly still not completely clear to me why diffusion is the optimal choice here, as opposed to alternatives and the single comparison to a CGAN seems unsatisfactory in terms of alternatives-exploration.

**Questions:**

I've interspersed some questions in the weakness section above. My main question has to do with the generalization/robustness of Haland. Does the natural language component of Haland _actually_ afford generalization to truly novel human partners? Because from what I can tell, the only "novelty" in the test set is a difference in how the same (semantically speaking) tasks are expressed, i.e. we are simply paraphrasing the instructions. The fine-tuning of the translator that happens during training worries me as potentially overfitting the language understanding to the predefined set of BRs the method was trained on. This is further exacerbated by the human designed TL which seems to incorporate some task-specific (e.g. Overcooked-specific) elements. I worry that these factors essentially eliminate the true generality of language, and the only robustness to variability we obtain is to paraphrasings. If this is the case, what is the point for natural language in this set up? Couldn't the human partners simply communicate their intent by selecting it from e.g. a menu?

Or is the assertion that the diffusion process which compresses BRs into some latent space will lead to generalization to any BR? I am not fully convinced that this was demonstrated.

---

### Official Review · Reviewer_zy87 · 2024-11-01

**Soundness:** 2
**Presentation:** 3
**Contribution:** 2
**Rating:** 5
**Confidence:** 3

**Summary:**

This paper introduces HALAND, a novel method utilizing diffusion models to estimate the policy networks of agents that cooperate with humans or human-like agents by processing language-based instructions. Experiments on Overcooked and Assistive Gym demonstrate that HALAND’s policies outperform those generated by baseline MLP models and its own ablations.


HALAND Method Summary: Each policy is represented as an MLP network. The authors use a VAE to encode the MLP policy parameters into latent representations, and the diffusion model estimates these latent representations. To ensure the agent follows human instructions, the authors implemented a translator, which first encodes language instructions with BERT and then further encodes them using the VAE. These encoded instructions are used as conditions for the diffusion model, facilitating agent alignment with human input.

**Strengths:**

This paper provides strong qualitative and quantitative results, along with diverse ablation studies that demonstrate HALAND’s effectiveness on Overcook and Assistive Gym tasks. The ablations also confirm the suitability of the chosen diffusion backbone for this approach.

**Weaknesses:**

The approach of using diffusion models to generate network parameters is not very convincing, especially given the existence of diffusion policy works that generate action sequences effectively. The paper lacks a baseline capable of directly handling natural language inputs, which weakens the soundness of experiments. Furthermore, in tasks requiring cooperation through natural language, recent studies have widely explored using large language models (LLMs) to generate actions; however, this paper lacks relevant discussion or experimentation on this approach. The details can be found in the questions section.

**Questions:**

1. **Comparison with BERT-Augmented MLP**: I recommend that the authors consider a baseline that concatenates BERT representations with the MLP inputs, either as a new baseline or as part of an ablation study.

 *Reason of this question*: Since this study uses natural language as task instructions, it is essential to evaluate policies that integrate language models like BERT directly into the MLP policy, as simple MLP models cannot interpret language instructions without these representations. Ablation studies indicate that the method collapses without the translator (BERT+VAE), suggesting that language instructions play a crucial role. Including BERT in the baselines would help demonstrate the effectiveness of HALAND’s generative model for policy parameter estimation over direct conditioning on NL inputs.

2. **Diffusion Policy for Action Generation**: This paper uses diffusion models to generate policy network parameters, which could be viewed as simulating MLP optimization. However, prior research [1] has applied diffusion models to generate action distributions, directly aligning with diffusion models’ denoising functions. The authors should discuss this line of work and clarify why diffusion models are used here to generate policy parameters instead of actions.

3. **Generalizability to Complex Policy Networks**: A potential limitation of HALAND is its scalability to more complex policy architectures, which may present challenges for diffusion models in terms of parameter estimation. How could HALAND extend to larger policy models with significantly more parameters?

4. **Large Language Models for Cooperative Agents**: For agents that take natural language as input, using large language models (LLMs) for decision-making is an established and effective approach. There are relevant studies applying LLMs on tasks like Overcooked [2] and embodied AI [3]. It would strengthen the paper to include a discussion of these methods or a simple baseline using an LLM (such as GPT-4) for comparison.

5. **Performance Over Oracle**: It would be useful to explain why HALAND outperforms the Oracle in some partner setups, since the authors state that Oracle are the best responses.

Minor Issues:
The word “BERT” should be consistently capitalized.

[1] Diffusion Policy: Visuomotor Policy Learning via Action Diffusion. Cheng Chi et al., RSS’23.

[2] ProAgent: Building Proactive Cooperative Agents with Large Language Models. Ceyao Zhang et al., AAAI’24.

[3] Building Cooperative Embodied Agents Modularly with Large Language Models. Hongxin Zhang et al., ICLR’24.

---

### Official Review · Reviewer_hyVL · 2024-11-02

**Soundness:** 2
**Presentation:** 2
**Contribution:** 2
**Rating:** 5
**Confidence:** 2

**Summary:**

The work presents Haland, an approach for training a cooperative agent for human-AI coordination. Haland first trains several best-response policies, then compresses the policies into a single language-conditioned diffusion model. The diffusion model offers expressivity of the multiple modes in best-response policies that are able to be expressed within the diffusion model. During training, a set of different human partner policies paired with natural language instructions are provided. A VAE compresses the policy parameters. Then, the latent space of the VAE is used to train the conditional diffusion model.
 The authors empirically evaluate across different cooperative environments and show that partners can give natural language instructions to get improved zero-shot coordination.

**Strengths:**

The results are promising in that they show Haland achieves the best overall coordination performance on all benchmarks and is comparable to an oracle policy.

**Weaknesses:**

There is no indication of how the model would handle discrepancies or novel instructions during test time, which raises concerns about its robustness in open-vocabulary dialogue situations.

It's unclear if the natural language instructions fully capture the dynamics of policy execution across different task states or moments. Instructions might only describe initial task actions or provide general summaries, potentially missing context-specific nuances.

The experimental setup lacks sufficient detail on the baselines used. More detail in the human evaluation would also be helpful.

**Questions:**

In open-vocabulary dialogue, the L_N natural language instruction space is extremely large, and could involve language instructions not seen at training time. How would the approach be affected by discrepancies or out-of-distribution natural language at test-time?

How generalizable is the assumption that the set of natural language instructions for each partner policy should share similar semantic meanings and differ only in expressions? Is it possible that people could interpret the same partner policy in completely different ways, fixating on different aspects of the partner policy, leading to semantically varied interpretations?

Are the natural language instructions encompassing of the overall policy rollout? Or do they contain temporal or state-based associations. For example, the natural language may describe what the policy does early on in the task, and then summarize actions in a different state in the task.

I’m finding myself needing a bit more context on the baselines used in the experiments. It would be helpful to add more detail on the baselines and connect them to approaches in prior work. Are any of the baselines implementations of prior approaches? Fictitious co-play (Strause et al.) and other SOTA population-based training baselines should be used, especially with respect to benchmark tasks, like Overcooked. Can you add Instructed-Adaptive Ego as well?

I’d like more detail on the human evaluation. What was the distribution of their natural language instructions? No statistical testing is conducted, so the results should be qualified in that they only point at trends. Was the study IRB approved?

**Details Of Ethics Concerns:**

The human evaluation falls under, to my understanding, a human-subjects study, asking participants questions regarding preference and experience with the trained agent. No statistical testing is performed. IRB approval should be sought or reported in the study.

---

### Official Review · Reviewer_fR1v · 2024-11-03

**Soundness:** 2
**Presentation:** 2
**Contribution:** 2
**Rating:** 3
**Confidence:** 4

**Summary:**

1. The paper introduces a novel framework, named Human-AI Coordination via Policy Generation from Language-guided Diffusion (acronym: Haland).

2. It compresses diverse best response policies into a single diffusion-based generator.

3. The effectiveness of Haland has been validated through empirical evaluations conducted across diverse cooperative environments. The intuitive nature of human expression has been harnessed through natural language instructions in order to tackle the challenge of zero-shot human-AI coordination during deployment.

4. Haland enhances policy deployment through language instruction, moving away from few-shot adaptation.

**Strengths:**

This paper handles the challenge of zero-shot human-AI coordination during deployment, harnessing the intuitive nature of human expression through natural language instructions.

The effectiveness of Haland framework has been validated through empirical evaluations conducted across diverse cooperative environments.

**Weaknesses:**

Line numbers in the paper have been quoted followed by an observation on weakness.

1. L024: the alignment between task-relevant and natural languages is achieved to facilitate ….

Observation: Please elaborate what is a task-relevant language? Shouldn't natural languages be task relevant too given user goals. The motivation for alignment is not clear when the definition of task-relevant language and it's difference from natural languages have not been described

2. L025: Empirical evaluations across diverse cooperative environments ….

Observation: Evaluations in Comparative and Mixed motive environments have not been carried out.

3. L028: and outperforms existing methods by approximately 89.64%

Observation: Please substantiate this result that you have mentioned in the abstract as the paper does not provide any results showing an approximate outperformance by 89.64%. Either the corresponding results are missing or the writing has to be improved significantly

4. L107, 109, 111 …. .Literature (Chaplot et al.,2018) combines

Observation: The readability needs to be improved. \citep has been used for citation with a redundant reference to literature. Better readability can be ensured with \citet latex command which would read as, "Chaplot et al et al., 2018 combines …."

5. L132-L134: .The remarkable success of Diffusion model in various domains has showcased its powerful generation capability and has been used in RL for planning or functioning as expressive policies recently (Yang et al.,2023).

Observation: Can the use of Diffusion mode in the present work be considered as a novel contribution if Yang et al covers this? Please share how your work is different from theirs

6. L207 – L211: To benefit from the both directions while overcoming the limitations, we propose to distill the multiple best response policies into a single NL guided diffusion model for policy generation, due to its powerful generation capability. Similar approach was first
proposed in literature (Hegde et al., 2023),which distils the quality diversity policy archive into the diffusion model conditioning on behavior descriptions.

Observation: In that case, what is the contribution of the present work of similar approach has been adopted by Hedge et al?

7. L298: Conversion to TL embedding in Haland may not be unique.

Observation: How can this non-unique conversion be handled?

**Questions:**

1. L017 – L018: …. which is unbearable in real-world applications such as health care and
autonomous driving ….

What do you mean by ‘unbearable’? Is this any engineering stress-test based observation? Please describe your understanding of "unbearable"

2. L145 and L308:  Please revise the phrasing

3. L254-L255: Second, natural language instructions often contain syntactic components irrelevant to specific tasks.

Give examples of components irrelevant to tasks.

4. L258:

Please fix typographic error of develope to developed

Please fix typographic errors in L356, L460, L488, L880, L1113 of HALAN to HALAND

---

### Official Review · Reviewer_nWBu · 2024-11-04

**Soundness:** 3
**Presentation:** 2
**Contribution:** 2
**Rating:** 6
**Confidence:** 4

**Summary:**

The paper introduces HALAN, a zero-shot Human-AI Coordination framework that distills several best response policies into one Language-guided Diffusion model. To produce cooperative policies in test-time, HALAN first translates user-provided natural language instructions into task-specific language representations with fine-tuned bert and Variational Autoencoder (VAE) and then models them using a latent diffusion model. The framework is evaluated on three two-agent cooperative benchmarks, with human studies and ablation studies demonstrating HALAN's effectiveness.

**Strengths:**

The approach is notable for consolidating various best-response policies into a single diffusion-based generator, which is an innovative and valuable contribution. The experimental results are compelling, and the ablation study covering individual components is particularly appreciated.

**Weaknesses:**

- Clarity of Writing: Some sections would benefit from clearer language. For example
   - line 279: "Bert model is encapsulated into a classifier," is confusing
   - Figure 5 could be clarified with added annotations of ego agent or partner agent to assist reader comprehension.

- The design choice of translating NL into TL: Unlike most language-conditional diffusion models, HALAN incorporates a step where natural language is translated into task-specific language, which will introduce the effort of designing this language for different tasks, and impact the potential generalization ability to different tasks. The ablation study of “W/o Translator” could use more depth and clarification. Specifically:
    - How is the diffusion model implemented to condition directly on natural language instructions?
    - Could the use of stronger text encoders, like T5-XXL, potentially mitigate the gains of using this task-specific translator?

- The policy backbone is rather simple compared to the latent diffusion model incorporated in the proposed method. Whether the baselines would also benefit from a comparable policy backbones remains unclear.

**Questions:**

Please see the weaknesses.

---

### Meta-Review · Area_Chair_nsX4 · 2024-12-11

**Metareview:**

This paper introduces HALAND, a framework for zero-shot human-AI coordination that leverages natural language instructions to guide policy generation. The method compresses multiple best-response (BR) policies, trained via reinforcement learning for diverse human partners, into a language-conditioned diffusion model. A translator module, built on BERT and a Variational Autoencoder (VAE), converts natural language instructions into a structured "task language" representation, which is then used to condition the diffusion model during both training and inference. HALAND is evaluated across three cooperative environments—Overcooked, Level-Based Foraging, and Assistive Gym.

Most reviewers appreciated the zero-shot human-AI motivation, significant empirical evaluations, and strong results. Reviewers raised concerns about clarity in the method and noted that the writing needs improvement. They mentioned that the experimental section lacks sufficient detail, particularly regarding the baselines. Reviewers also requested more details about the human evaluation and raised questions about leveraging large language models for natural language tasks. Additionally, they questioned generalization to novel instructions and scalability to models with higher parameters, among other points.

The AC concurs with the majority of reviewers and notes that the concerns raised remain unresolved due to no response from the authors. The feedback from five reviews includes constructive suggestions to improve clarity, experimentation, and scope of this manuscript for a potential resubmission.

**Additional Comments On Reviewer Discussion:**

No rebuttal was submitted by the authors. During the reviewer discussion phase, reviewer nWBu was asked about potentially championing the paper, but that wasn't supported.

---

### Decision · Program_Chairs · 2025-01-22

Reject